# Development of Analog Rice Made from Cassava and Banana with the Addition of Katuk Leaf (*Sauropus androgynous* L. Merr.) and Soy Lecithin for Lactating Women

**DOI:** 10.3390/foods13101438

**Published:** 2024-05-07

**Authors:** Meta Mahendradatta, Esra Assa, Jumriah Langkong, Abu Bakar Tawali, Dwi Ghina Nadhifa

**Affiliations:** Department of Agricultural Technology, Faculty of Agriculture, Hasanuddin University, Makassar 90245, Indonesia; esraassakalapadang@gmail.com (E.A.); jumriah_langkong@yahoo.com (J.L.); abubakar_tawali@yahoo.com (A.B.T.); dgnadhifa@gmail.com (D.G.N.)

**Keywords:** analog rice, *Sauropus androgynous*, lactating women, banana, cassava

## Abstract

The development of analog rice, apart from being an effort to diversify food, also has the potential to be developed as a functional food to fulfill the nutrients needed by a community. Katuk leaf is known for its ability to accelerate the breast milk production of lactating women, which is inseparable from sterol in terms of bioactive content. This study aimed to determine the best formulation of analog rice made from cassava flour, banana flour, Katuk leaf powder, and soy lecithin that was sensorily acceptable, in a shape resembling rice, and able to fulfill the nutritional needs of lactating women. Analog rice was produced using an extruder machine before the physical and sensory properties analyses were carried out, followed by the chemical properties analysis. Formulation C (80% cassava flour, 20% banana flour, 3% Katuk leaf powder, and 0.5% soy lecithin) was obtained as the best or most preferred formulation based on sensory analysis. The resulting grain was oval–round in shape and had a green-brownish color, fluffy texture, and distinct aroma and taste derived from the raw material used. Therefore, this research is expected to support the development of analog rice for providing the main staple food to fulfill lactating women’s nutrition.

## 1. Introduction

Analog rice is a rice-shaped material processed from a mixture of rice and non-rice flours, either with identical or better nutritional content than paddy rice [1]. Rice consumption as a staple food is rising in Indonesia following population growth, demanding more rice imports [2,3]. Analog rice has been identified as a potential substitute for paddy rice that will help overcome issues with food availability and can also be designed to impart specific nutrients needed [2,4]. One of the groups of people who need certain nutrients is lactating women. Insufficient breast milk production is a common issue that many lactating women face [5,6,7]. Katuk leaves can help increase breast milk supply for lactating women due to their active compounds, alkaloids, and sterols [8,9]. The addition of Katuk leaves when making analog rice is carried out in an effort to fulfill the nutritional needs of lactating women.

Analog rice can be made from local carbohydrate sources, such as tubers, cereals, pulses, etc., the choice of which will determine its nutritional composition and specific characteristics [10,11,12]. Among the local carbohydrate sources that can be utilized are cassava and bananas. Nutritionally, cassava is rich in carbohydrates, calories, vitamins, minerals, and protein, as well as various types of fiber [13,14,15]. The amount of carbohydrates contained in cassava is higher than that in other types of carbohydrate sources that are commonly consumed, such as rice and corn which are 40% and 20% lower than cassava, respectively [14,16]. Furthermore, bananas are a nutritious alternative to rice due to their high carbohydrate content [17,18]. Besides the main ingredients, additional ingredients are necessary to reduce rehydration time and obtain firm, non-breakable rice grains [19]. Soy lecithin can be added to analog rice extrusion products to improve texture, reduce adhesive power, and improve the shape of the final product after the hydration process [20]. Soy lecithin is also beneficial in increasing milk production for lactating women due to its choline component, which prevents plugged ducts without any unpleasant side effects [21,22]. This shows the potential of both cassava and banana to be alternatives to rice, which has an important role in supporting food security and diversification through the development of analog rice with the addition of soy lecithin to improve its texture.

The development of analog rice by utilizing local ingredients has previously been carried out. Sada et al. used mocaf, mung bean, and purple corn to make analog rice [23], Rumitasari combined white corn and mung bean [24], Hasbullah et al. used mocaf flour and modified suweg flour [25], while Sulfi et al. made analog rice consisting of cassava and mung bean [26]. The production of analog rice with a combination of cassava flour and banana flour, with the inclusion of Katuk leaf powder and soy lecithin, has never been attempted before where the analog rice produced has additional nutritional value and serves as a functional food, particularly for lactating women. Lactating women need 500–1000 more calories than non-breastfeeding women [27].

Based on this description, it is necessary to conduct further research regarding the proper formulation in the production of analog rice derived from local Indonesian ingredients to replace rice and be able to meet the nutritional needs of lactating women with an acceptable taste. Therefore, this study aims to obtain the best formulation of analog rice based on the preferences of nursing mothers, to produce analog rice that resembles regular rice, and to analyze its physical and chemical properties.

## 2. Materials and Methods

### 2.1. Materials

The materials used in this study consisted of cassava flour (*Manihot esculenta*) (supplied by “Kusuka” Ubiku, Bantul, Yogyakarta, Indonesia), banana flour (*Musa paradisiaca* L.) (supplied by Lingkar Organik, Yogyakarta, Indonesia), Katuk leaf powder (*Sauropus androgynus* L.) (supplied by Herbadream, Solo, Indonesia), and soy lecithin (supplied by Buana Chem, Bandung, Indonesia). All chemicals used were of the analytical grade (Pro Analyst Grade; Merck, Rahway, NJ, USA).

### 2.2. Production of Analog Rice

The analog rice formulation refers to Yulianti and Waluyo [17] with modifications on flour composition and proportions through the addition of Katuk leaf powder and soy lecithin to provide the desired nutrition needed for lactating women. The formulation for making analog rice can be seen in Table 1. The chosen concentration of Katuk leaf powder was 3%, which was obtained from preliminary research, while the chosen concentration range of soy lecithin was predicated by referring to Oke et al.’s study [28].

Analog rice was produced using extrusion technology, which consists of several process stages (Figure 1) including material preparation, mixing, steaming, molding, and drying. The preparation of the material began with sifting the Katuk leaf powder using a 100-mesh sieve. Furthermore, the ingredients, namely cassava flour, banana flour, Katuk leaf powder, and soybean lecithin, were weighed according to the formulation. The next stage was the mixing process. The ingredients, consisting of flour and soy lecithin, were mixed first and then stirred for 3 min until evenly distributed. Then, 20% water was added little by little to the ingredients that had been mixed and stirred again for 10 min until the ingredients were evenly mixed and formed a dough with a slightly wet texture. Next, the dough was preconditioned by wrapping it in a filter cloth and steaming it in the boiler for 30 min. After that, it was extruded to form analog rice grains using a single screw extruder machine, assembled in the CV Giat Extruder Machine, Bogor, West Java, Indonesia. During the extrusion process, the dough flowed and was formed through the die (mold). The obtained analog rice grains were then dried in a blower oven (temperature 60 °C) to reduce the moisture content of the analog rice to below 14% (wet basis moisture content).

### 2.3. Physical Analyses of Analog Rice

#### 2.3.1. Bulk Density

Bulk density was measured by placing the sample in a measuring cup to a certain volume without compacting it, and then weighing the sample [29]. Bulk density was analyzed according to the following equation:$$\mathrm{Bulk}\;\mathrm{Density}\;(g/\mathrm{mL})=\frac{Sample\;Weight\;(g)}{Sample\;Volume\;(mL)}$$

#### 2.3.2. Water Absorption

Water absorption was measured as prescribed by Yudanti and Waluyo [17]. A total of 1 g of sample was weighed. It was then put into an empty centrifuge tube that had been weighed before, and 10 mL of distilled water was added. Next, the samples were shaken using a vortex for 5 min until evenly mixed, then centrifuged for 15 min at 2000 rpm. After that, the supernatant was separated, and the centrifuge tube was weighed again. The value of water absorption was analyzed according to the following equation:$$\mathrm{WA}\;(\%)=\frac{Final\;sample\;weight\;\left(g\right)\;-\;Initial\;sample\;weight\;(g)}{Final\;sample\;weight\;(g)}\;\times \;100\%$$

#### 2.3.3. Swelling Power

Swelling power was measured by referring to Yudanti and Waluyo [17]. A sample of 5 g was weighed, and the sample that was weighed was randomly taken from as many as 10 grains. Then the diameter of the sample was measured in three orientations (top, side, front). Next, the sample was immersed, and its diameter was calculated again from the three-grain sides (top, side, and front) using a digital caliper. Swelling power was calculated using the following equation:$$\mathrm{Measurement}\;\mathrm{of}\;\mathrm{analog}\;\mathrm{rice}\;\mathrm{grain}\;\mathrm{diameter}\;(Ø)=\frac{\left(Ø1+Ø2+Ø3\right)}{3}$$
$$\mathrm{Swelling}\;\mathrm{Power}\;(\%)=\frac{ØB-ØA}{ØA}\;\times \;100\%$$where ØA is the diameter of analog rice before immersion (mm); ØB is the diameter of analog rice after immersion (mm).

#### 2.3.4. Cooking Time

Analog rice was cooked in a conventional method using a traditional cooker or steamer [30]. The method of cooking the analog rice with a traditional rice cooker began with a sample of 50 g being weighed. Then, the rice cooker was filled with 300 mL of water and heated on the stove for ±3 min until the water boiled. Next, analog rice was put into the boiler and cooked for ±4 min. If the analog rice was half cooked, then the analog rice was removed, and the remaining cooking water was replaced with new water. The rice cooker was then filled with 700 mL of water again, and the analog rice was steamed for ±10 min until it turned into rice, which was characterized by having no white spots in the middle and the rice texture turning chewy. The cooking time was calculated from the time the analog rice was put into the rice cooker until it was cooked.

#### 2.3.5. Yield of Cooked Analog Rice

The calculation of the yield of analog rice was determined from the comparison of the final weight of analog rice after drying with the initial weight of analog rice obtained from the weight of the raw material used [31]. The yield value was analyzed according to the following equation:$$\mathrm{Yield}\;(\%)=\frac{Analog\;rice’s\;final\;weight}{Analog\;rice’s\;initial\;weight}\;\times \;100$$

### 2.4. Chemical Analyses of Analog Rice

#### 2.4.1. Proximate Analyses

Proximate analyses carried out in this study consist of moisture, ash, protein, fat, and carbohydrate content by referring to the Association of Official Analytical Chemists (AOAC, 2006) [32]. Moisture and ash content were determined based on the gravimetric method on a wet basis, protein content determination was based on Kjeldahl analysis, the total fat was determined using Soxhlet extraction, and carbohydrate content was determined using carbohydrate by difference.

#### 2.4.2. Crude Fiber

The crude fiber of the analog rice was determined based on the SNI method [33]. Briefly, a total of 2–4 g of the sample was weighed and defatted using the Soxhlet method or by stirring in an organic solvent three times. The sample was dried and put into an Erlenmeyer flask, and then boiled after 1.25% H_2_SO_4_ was added followed by the addition of 3.25% NaOH, and it was then reboiled for 30 min. After that, the sample was filtered and the precipitate contained in the filter paper was washed with hot H_2_SO_4_, hot water, and ethanol. Subsequently, the filter paper was dried then cooled and weighed until a constant weight was achieved. Crude fiber content was calculated using the equation as shown below:$$\mathrm{Crude}\;\mathrm{fiber}\;(\%)=\frac{B}{A}\;\times \;100\%$$
If crude fiber > 1% then the equation is as follows:$$\mathrm{Crude}\;\mathrm{fiber}\;(\%)=\frac{C}{A-B}\;\times \;100\%$$
where A is the sample weight (g); B is the weight of precipitate in filter paper (g); C is the ash weight (g).

#### 2.4.3. Total Calories

The total calories in food can be calculated indirectly, based on the carbohydrate, protein, and fat content [34]. A total of 1 g of carbohydrate is equivalent to 4 calories, 1 g of protein is equivalent to 4 calories, and 1 g of fat is equivalent to 9 calories. Energy values are expressed in kilocalories (Kcal). The calculation of total calories can be obtained with the following equation:Calories (Kcal) = Protein × 4 + Carbohydrate × 4 + Fat × 9

#### 2.4.4. Phytosterol Levels

Phytosterol levels, namely sitosterol and stigmasterol, were determined according to the method prescribed by Indrayanto et al. [35]. Samples were weighed to within ±0.25 g and put into a 25 mL volumetric flask. Next, alcohol was added to as much as 1/3 of the volumetric flask, and the flask contents were then homogenized for 2 h. The sample was then filtered, and the filtrate obtained as a result of the filtering was spotted onto the TLC plate as many as 5 μg. After that, it was extracted (eluted) with the eluent CHCl_3_: Ethanol: Ethyl Acetate for ±45 min, then measured with a TLC scanner with a wavelength (λ) = 285 nm for sitosterol analysis and a wavelength (λ) = 264 nm for stigmasterol analysis.

#### 2.4.5. Total Flavonoids

The total flavonoids of the analog rice were determined according to Sri Widyawati et al. The sample was weighed to a maximum of 0.05 g, then dissolved in 10 mL of ethanol, and then filtered, and the filtered sample was diluted. After that, a standard quercetin solution was made by weighing 0.01 g of quercetin and diluting it to 1000 ppm. Dilution was carried out by pipetting 1 mL of 1000 ppm quercetin and adding 9 mL of ethanol to obtain 100 rpm quercetin. The standard series was prepared by pipetting 100 mL of quercetin to volumes of as much as 0.1 mL, 0.2 mL, 0.4 mL, 0.8 mL, and 1.6 mL, respectively. Then, each volume was made up to 5 mL with the addition of ethanol to obtain quercetin in a row at 2 ppm, 4 ppm, 8 ppm, 16 ppm, and 32 ppm. Subsequently, the flavonoid test was carried out, with a sample of 0.5 mL added to 3 mL of methanol, then 0.2 mL of 10% AlCl_3_, and 0.2 mL of CH_3_COOK 1 M. Furthermore, the volume was made up to 10 mL by adding 6.2 mL of distilled water. After that, the absorbance was measured at its maximum wavelength (400–500 nm) using a UV–Vis spectrophotometer. Quercetin was used as a standard, while ethanol and methanol were used as blanks [36].

#### 2.4.6. Iron (Fe) Levels

Iron levels were determined according to Nasution et al. Before preparing the sample, it was dried in an oven to reduce its water content, followed by ashing. The sample was prepared by adding 6 mL of 65% HNO_3_ and 2 mL of 30% H_2_O_2_ into a porcelain cup. Next, the results of the ash content were transferred to a 50 mL beaker until there was no residue left in the porcelain cup (the addition of 65% HNO_3_ was adjusted until there was no residue left in the porcelain cup). After that, the beaker containing the sample was heated using a hotplate until the solution was reduced by half (the destruction process). Then, 10 mL of distilled water was added, and it was filtered into a 50 mL volumetric flask, then squeezed up to the mark of the flask using distilled water and homogenized. After the sample was prepared, it was then injected by placing it in a cuvette, and the injection process was carried out using an AAS type AA-7000 Shimadzu. After that, if the sample was measured and exceeded the standard limit, a dilution process was needed so that the dilution factor could be obtained and included in the calculation. During sample injection, the standard solution series and the Fe wavelength were marked on the AAS instrument [37].

### 2.5. Sensory Analysis of Analog Rice

The sensory analysis used in this study was the hedonic method to determine the best formulation based on panelists’ preference with three levels of scale such as dislike (1), neutral (2), and like (3). The panelists were 25 breastfeeding mothers aged 18–40 years who were asked to observe their preferences for the cooked analog rice, which included color, odor, texture, taste, and overall appearance.

### 2.6. Statistical Analysis

The resulting data were processed using a completely randomized design (CRD) with three repetitions, and the results obtained for each parameter were analyzed using an ANOVA with three repetitions. Duncan’s test was used to confirm the differences between treatments. The software used for data processing was Microsoft Excel 2016 and IBM SPSS Statistics 24.

## 3. Results and Discussion

### 3.1. Analog Rice

The manufacture of the analog rice in this study used hot extrusion technology which, in its processing, used temperatures above 70 °C obtained from steamed heaters [38]. The preferred analog rice produced in this study (Figure 2) has a shape resembling rice in general, which is oval with a brownish-green color derived from the raw materials of the cassava, banana, and Katuk leaves used. After cooking, the analog rice has a fluffy texture like rice in general, and a deeper color, as well as aroma, and the taste produced has distinctive characteristics derived from the ingredients used. Analog rice can be cooked using an electrical rice cooker or the conventional method with a steamer. In this research, the analog rice was cooked using the conventional cooking method. The cooking of analog rice using a steamer can be performed by filling the steamer pot with water up to the limit of tera and bringing it to a boil, then, the analog rice is put in and boiled until half cooked, after which the steamer pot filter is removed and the remaining water from cooking the analog rice is replaced with new water then the analog rice is steamed. The cooking of analog rice lasts for 10–15 min and the cooked rice is characterized by not having white spots in the middle and having a chewy texture [30].

### 3.2. Physical Properties

The analog rice formulations produced had no significant effect (*p* > 0.05) on all of the physical parameters consisting of bulk density, water absorption, swelling power, cooking time, and yield of the analog rice, which can be seen in Table 2. The bulk density obtained in all three analog rice formulations showed no significant differences, ranging from 0.55 to 0.57 g/mL, with a ratio of cassava and banana flour of 60%:40% as the highest. Analog rice with a greater bulk density indicates that the porosity of the analog rice is lower [39,40,41]. This is attributed to water loss during the drying process in analog rice-making [40,42,43] and is influenced by the moisture content of the ingredients [44,45]. A large bulk density value will require a smaller storage space, and vice versa [43,46]. Bulk density is also influenced by the addition of soy lecithin which, according to Hartono et al., contains more than 90% fat, which has a lower density [47] resulting in a decrease in bulk density [47,48,49].

Generally, the water absorption capacity of a material is related to its bulk density value [41,45]. Low bulk density in analog rice indicates that the analog rice has high porosity [41,50]. According to Yulviatun et al., the higher the porosity of the analog rice produced, the greater the water absorption due to the greater number of cavities between particles [41]. The highest water absorption value of the analog rice was obtained from the sample with an 80%:20% ratio of cassava and banana flours (77.27%), with no significant difference from the other two formulas. The high water absorption was influenced by the high starch content of the ingredients used in the formulation, especially the amylose content which has an amorphous region [51,52,53]. In addition, soy lecithin, as a stabilizer, can absorb water and increase the water absorption of analog rice because soy lecithin has hydrophilic groups so its ability to bind water from the air will be faster [54,55,56].

The highest swelling power value was obtained in the 80%:20% ratio of cassava and banana flour (31.64%), which was influenced by the amylopectin and amylose content contained in the ingredients. Banana has a low amylose content, which is 11.2% [57], while cassava has a higher amylose content, around 30% [58]. The high starch and amylose content found in the ingredients affects the swelling power [59]. Amylose, which has amorphous regions, is reactive to water molecules, causing the amount of water absorbed into the starch to increase [60,61]. Furthermore, the addition of soy lecithin affected the swelling power of the analog rice. Phospholipids present in soy lecithin can easily associate with the starch present in the material to form starch–lipid complexes, and these complexes limit the swelling of starch granules [59,62,63].

Cooking time shows the length of time needed to cook the rice until it is cooked. The results of the cooking times of the analog rice made from cassava flour and banana flour from the three formulations were around 14 min. The analog rice cooking in this study was performed conventionally using a steamer pot with boiling and steaming stages. The time needed to cook the analog rice in this study was 6 min faster when compared to paddy rice, which took 20 min. Herawati et al. stated that rice requires a longer cooking time of 20.5 min [64], while the cooking time required for analog rice is 10–15 min faster than paddy rice [65]. This is influenced by the process of making analog rice that has been through a pre-heating process so that the analog rice starch has been gelatinized [38,66]. As a result, the water absorption of the starch increases and the cooking time becomes shorter [67]. The higher the water absorption and cooking temperature, the faster the cooking time [68,69].

The yield obtained in the production of the analog rice indicates the loss of product during the process. The highest yield was obtained in the analog rice formulation with a ratio of cassava flour and banana flour of 80%:20% (67.67%), followed by the 70%:30% ratio (61.73%), and the 60%:40% ratio (58.58%). This was influenced by the addition of soy lecithin to the formulation which functions as a binder to improve the texture and improve the shape of the final product after the rehydration process, and to reduce adhesion [20,70], increasing yield. This follows the findings of Aini et al. and Van Buren et al., which state that binders function to reduce cooking loss during the processing process and cause product yields to increase [71,72]. In addition, the yield value can be influenced by temperature, drying time, and moisture content [73,74].

### 3.3. Proximate Content

The results obtained, as shown in Table 3, showed that the differences in the three analog rice formulations had no significant effect (*p* > 0.05) on the moisture content, ash content, protein content, and carbohydrate content of the analog rice, but had a significant effect (*p* < 0.05) on the fat content of the analog rice produced. The moisture content of the three analog rice formulations (around 7%) met the requirements of safe moisture content for rice according to Indonesian National Standard, which is <14% [75]. A moisture content of <14% will prevent mold growth that often occurs on rice, cereals, and grains [30,76]. According to Mishra et al., analog rice needs to be dried to a moisture content of 4–15% to achieve optimal moisture content to increase shelf life [77]. The moisture content of a food product can be influenced by the addition of water to the ingredients [38], the steaming process [78,79], and the drying process [78,80].

Ash content is the resulting ash that remains from a sample of food that is completely burned in the ashing process. The ash content obtained from the three analog rice formulations (1.54–2.32%) complied with the requirements for analysis of ash content in healthy foods according to Indonesian National Standard, namely, 3.50% maximum [81]. According to The Indonesian Food and Drug Authority, an ash content below 3.50% is good for consumption because the minerals contained in it are good for health [82]. The ash content in food can be influenced by the type of material used [83,84] and the drying process [85,86]. The drying process results in the decomposition of the water molecule bonding components and also increases the mineral content, increasing the ash content [87,88]. However, the ash content in food decreases, which is potentially caused by the steaming process due to mineral solubility in water during the heating process, resulting in a lower ash content [89,90].

The protein content results had no significant difference and ranged from 2.76 to 3.07%. This result was lower than that of regular rice that is generally available, which is IR64 (7.18%) [91]. Despite soy lecithin containing a protein content of around 232–1338 mg/kg [92,93], the percentage of protein source addition in the production of analog rice is insufficient to elevate the analog rice’s protein content. A similar result was shown in Sumardiono et al.’s research, where the obtained protein content of analog rice ranged from 2.66 to 4.83% [91]. Extrusion and heating processes can reduce protein content. Heat application and extrusion in analog rice production break hydrogen bonds causing denaturation and structural modification which damage the protein structure leading to decreased protein content [50,94].

The results of the fat content analysis obtained in this study showed a low value when compared to rice in general, namely, 1.37%. As the concentration of cassava flour increases, the concentration of banana flour decreases, and the fat content of the analog rice produced decreases. The highest fat content was obtained at a ratio of 60%:40% cassava and banana flour (0.80%), followed by a ratio of 70%:30% (0.60%), and a ratio of 80%:20% (0.48%), with a significant (*p* < 0.05) difference. Fat content in analog rice production can function as a lubricant in the extruder machine to facilitate the extruding and molding of the dough. The extruder’s performance is impacted by high fat levels (above 5–6%), which can result in poorer cooking and product forming. However, low levels of fat (below 5%) can improve the texture and facilitate steady extrusion [95]. This is in line with Mamuaja et al.’s results in analog rice production consisting of purple sweet potato, banana, and sago which had a fat content ranging from 0.25 to 1.04% [18]. Analog rice with a low fat content is less susceptible to becoming rancid and has a longer shelf life.

The carbohydrate content obtained in the analog rice production was higher when compared to rice, which contained 80.14% carbohydrates [96]. The carbohydrate content of the three types of analog rice produced from cassava and banana flours was similar, ranging from 86 to 87%. The high carbohydrate content of the analog rice obtained indicates that the analog rice can be used as an alternative source of carbohydrates and calories to rice. Budi et al., in their research related to analog rice made from corn flour, sorghum flour, and starch, obtained a carbohydrate content of 91% [38], while Noviasari et al. produced analog rice with a combination of sorghum and mocaf flours with the addition of arenga starch and corn starch, and obtained a carbohydrate content of 91–94% [97]. Cassava flour and banana flour were used as carbohydrate sources, resulting in the production of analog rice that had a high carbohydrate content [98].

### 3.4. Total Calories

Calories can be interpreted as an energy unit that describes the amount of potential energy contained in a food. The results of the total calorie analysis can be seen in Figure 3. The results of the analysis of variance showed that the comparison of cassava flour and banana flour in the formulation of the analog rice had no significant effect (*p* > 0.05) on the total calories of the analog rice, with the rice with a ratio of 70%:30% having the highest value (367.36 Kcal). The high results obtained on total calories are influenced by the levels of carbohydrates, proteins, and fats contained in the ingredients used. The higher the amount of the three components, the higher the energy contributed, and, likewise, the total calories produced will be higher. Analog rice calories obtained in this study were around 362–367 Kcal, which was higher than regular rice, namely, 360 kcal [30]. The high total calories in analog rice can be used as energy intake in carrying out activities, especially for lactating women who require a higher calorie intake of 500–1000 [27]. The sample with a ratio of 80%:20% had the lowest caloric value, which could be influenced by the low fat and carbohydrate content; thus, the calculation resulted in a low total caloric gain. This is confirmed by Schriani & Yulianti, who say that the energy value of food is determined by calculating the composition of carbohydrates, fats, and proteins [99].

### 3.5. Crude Fiber Content

Crude fiber is a part of dietary fiber that cannot be hydrolyzed by certain chemicals, namely sulfuric acid (H_2_SO_4_) and NaOH. The results of the analysis for crude fiber content can be seen in Figure 4. The analog rice formulation, in comparison to cassava flour and banana flour, had no significant effect (*p* > 0.05) on the crude fiber content of the analog rice. The crude fiber content in this study, which ranged from 6.45 to 7.01%, was higher when compared to several rice varieties of Indonesia, which ranged from 0.43 to 1.83% [100]. This was possibly influenced by the ingredients used, which consisted of cassava flour, banana flour, and Katuk leaf powder. The crude fiber content found in cassava flour is 2.41% [101], and in banana flour is 2.0% [102], while Katuk leaf contains between 1.07 and 1.87% crude fiber [103]. The more additional ingredients used, the higher the analog rice’s crude fiber content would be. However, the heating process can impair the fiber component in food which leads to fiber degradation and can also change the ratio of soluble and insoluble fibers [104].

### 3.6. Phytosterol Levels

The phytosterol analyses carried out in this study consisted of determining the sitosterol and stigmasterol levels. The results of the phytosterol analyses can be seen in Figure 5. The results showed that the comparison of cassava flour and banana flour with the addition of soybean lecithin had no significant effect (*p* > 0.05) on the β-sitosterol levels of the analog rice, whereas it did have a significant effect (*p* < 0.05) on the stigmasterol levels of the analog rice. The increase in the phytosterol levels contained in the analog rice was potentially caused by the addition of Katuk leaf powder. Katuk leaf powder contains 2433.4 mg/100 g dry of phytosterol [105], which contributes to the sterol and stigmasterol levels in analog rice. In this study, the greater the addition of soy lecithin, the higher the sitosterol content of the analog rice obtained, ranging from 1.50 to 1.37%, with a ratio of 80%:20% as the highest. Sitosterol can be obtained from soybean lecithin, which is the residue from the processing of soybean oil and contains sitosterol. This is in accordance with Krisnawati, which indicates that soybean oil contains 300–400 mg of sterols per 100 g, with levels of β-sitosterol (53–56%), stigmasterol (17–21%), and campesterol (20–23%) [106]. This is also confirmed by the statement by Sihmawati & Rosida, that soy lecithin contains a sterol component of 2–5% [107].

The analog rice with a ratio of 60%:40% showed a high stigmasterol content (1.60%), and this was significantly (*p* < 0.05) higher than that of the analog rice with a 80%:20% ratio (1.11%). The decline in stigmasterol content in the analog rice is due to the reduced ratio of banana flour in the analog rice formulation. The study by Ramu et al. found that banana extract contains a fairly high stigmasterol content of 21.91% [108]. As a result, the lower the concentration of banana flour, the lower the stigmasterol content in the analog rice produced. In addition, Katuk leaves contain stigmasterol, which can be used to increase lactation in lactating women. According to Petrus, one of the stigmasterol components in Katuk leaves is stigmasta-5,22-dien-3β-ol [109], which functions in the same way as cholesterol in the process of steroidogenesis, namely, converting free cholesterol into pregnenolone (precursor of all hormones including those for producing breast milk) [110,111]. Furthermore, sterol has a function in modulating the gut microbiota (Figure 6), hence, it is beneficial for health and could improve breast milk quality [112]. β-sitosterol enhances the variety of *Staphylococcus* and *Streptococcus* bacteria found in the colostrum, bacterial species that are highly nourishing for infants [112,113].

### 3.7. Total Flavonoids

Flavonoids are phenolic compounds with the chemical structure C6-C3-C6 that are found in many plants and foods. The results of the analysis (Figure 7) showed that the different analog rice formulations had no significantly different effect (*p* > 0.05) on flavonoid content, with values ranging from 0.17 to 0.21%, with the highest value obtained from the rice with an 80%:20% ratio. Flavonoids can be obtained from the addition of Katuk leaves which contain flavonoids of the quercetin and kaempferol types. Those flavonoids affect the hormone prolactin, which works actively in the formation of breast milk so that milk production can run smoothly [114,115]. This was confirmed in the study by Magdalena et al., which found that the flavonoid type quercetin, with an amount of 4.5 mg and 138.14 mg of kaempferol, was found in Katuk leaves [116]. Moreover, the addition of soy lecithin to analog rice formulations increases the activity of flavonoids. Soy lecithin is a phospholipid that acts as a barrier against oxygen, thereby reducing the oxidation process or acting as an antioxidant during heating [117,118]. Furthermore, flavonoid levels in food are also influenced by the heating process, such as steaming and drying. Heating causes the flavonoids to easily oxidize, and phenol decomposition will occur, which will affect the flavonoid content [119,120,121].

### 3.8. Levels of Iron (Fe)

Iron is a macro mineral that is needed by the body and plays a role in the formation of red blood cells, especially in the synthesis of hemoglobin [122,123]. The results of the iron analysis (Figure 8) showed that the rice analog formulation consisting of different ratios of cassava flour and banana flour had no significant effect (*p* > 0.05) on iron levels, with the iron levels obtained ranging from 65.87 to 84.32 ppm, with the rice with a ratio of 70%:30% having the highest level. The results obtained indicate that the analog rice fulfills the nutritional iron needs of lactating mothers consuming at least 30 mg of iron [124]. Iron is needed by lactating women to fulfill nutritional adequacy in producing breast milk. Iron content in food is strongly influenced by environmental factors, both during processing and storage. According to Astuti et al., the stability of iron depends on several environmental factors, namely, exposure to air, light, and humidity, as well as the nature of the material [125]. In addition, analog rice processing consists of a steaming process that can also affect the iron content in the analog rice produced. Kusnadi stated that processing with traditional steaming can affect the nutritional content of analog rice, including the iron concentration [126]. The decrease in iron levels in foodstuffs during the boiling process was also shown from the results of Prasetyo et al., which indicated that there was a decrease in the iron content of beef liver and tempeh by 22.43–34.61% in the boiling process [127].

### 3.9. Sensory Analysis

Sensory analysis is a method of testing that uses the human senses as the primary tool for measuring consumer acceptance or evaluating product quality. Overall, as shown in the sensory radar chart (Figure 9), the best formulation of analog rice based on the panelist’s acceptance of four sensory parameters was obtained at a ratio of 80%:20% of cassava and banana flours. The dominant color produced by the analog rice was brownish-green. This was due to the addition of Katuk leaf powder and other ingredients. In general, Katuk leaves can be used as a natural green colorant and a source of calcium and protein [103,128]. The food color becomes fainter after cooking due to starch gelatinization which is influenced by cooking time and temperature [129].

Texture is an important food quality parameter for the acceptance of analog rice, and includes the rice’s fluffiness and stickiness. The results obtained indicate that the addition of soy lecithin made the texture of the analog rice produced more favorable, imbuing it with a fluffy texture. Soy lecithin, as an amphiphilic compound, has a hydrophilic group that can bind water to make the texture of food ingredients more stable [56]. According to Wang et al., lecithin as an emulsifier in analog rice functions to improve texture, reduce adhesiveness, and improve the shape of the final product after the hydration process [20].

Aroma is a sensation that is formed from the combination of the forming ingredients and their composition in a food ingredient captured from the sense of smell. The dominant aroma in the analog rice produced was due to the addition of Katuk leaf powder and other ingredients in the formulation, making the aroma present in the analog rice difficult to distinguish. The addition of Katuk leaf powder gave a distinctive aroma, namely a languorous aroma to the analog rice. The strong aroma in the flour is caused by the enzymes lipoxygenase and chlorophyllase, which are caused by cyanide compounds (HCN) [2,130]. The Katuk leaves gave a distinctive and pungent aroma even though their addition was in small amounts [131,132]. Meanwhile, another dominant aroma that appeared was the aroma of cassava flour as the main constituent ingredient in the analog rice produced.

Taste is formed from the combination of ingredients used in a product. The addition of several ingredients in making the analog rice made it difficult for panelists to distinguish the taste of the analog rice, as the taste was dominated by cassava as the main raw material used. Naknean and Meenune stated that the factors that affect food flavor are temperature, chemical compounds, concentration, and their interaction with other components [133]. While Kusmiandany reported that cassava gave a bitter taste due to its toxins, Arief et al., in their research, found that the flavor of analog rice made of cassava was almost the same as the original rice [134,135].

## 4. Conclusions

Based on the consumer preference level of lactating women, we concluded that the C formulation of analog rice was the best analog rice, containing 80% cassava flour, 20% banana flour, 3% Katuk leaves powder, and 0.5% soy lecithin. This analog rice sample had the highest level of protein, crude fiber, and total flavonoids and the lowest level of fat and calories. Furthermore, the preferable analog rice ratio exhibited a bulk density of 0.55 g/mL, 77.27% water absorption, 31.64% swelling power, and a cooking time of 14.07 min with a 67.67% yield. Formulations with a ratio of 80%:20% can be used as an alternative to original rice for lactating women who, by consuming the analog rice at around 300 g per day, can meet their calorie requirement, which is around 500–1000 kcal, as well as 20% of their protein requirement and 30–60% of their iron requirement. The analog rice produced had a shape resembling rice grains with oval characteristics. The sensory analysis showed a brownish-green color and a fluffy texture, as well as a distinctive aroma of Katuk and a distinctive taste of analog rice from the ingredients used. However, more efforts are still needed to improve the sensory quality of the analog rice produced to make it more acceptable to the public so that it can fully replace the available rice, and further studies need to be conducted on the functionality of the analog rice produced in order to investigate the efficacy of the analog rice in assisting lactating women.

## Figures and Tables

**Figure 1 foods-13-01438-f001:**
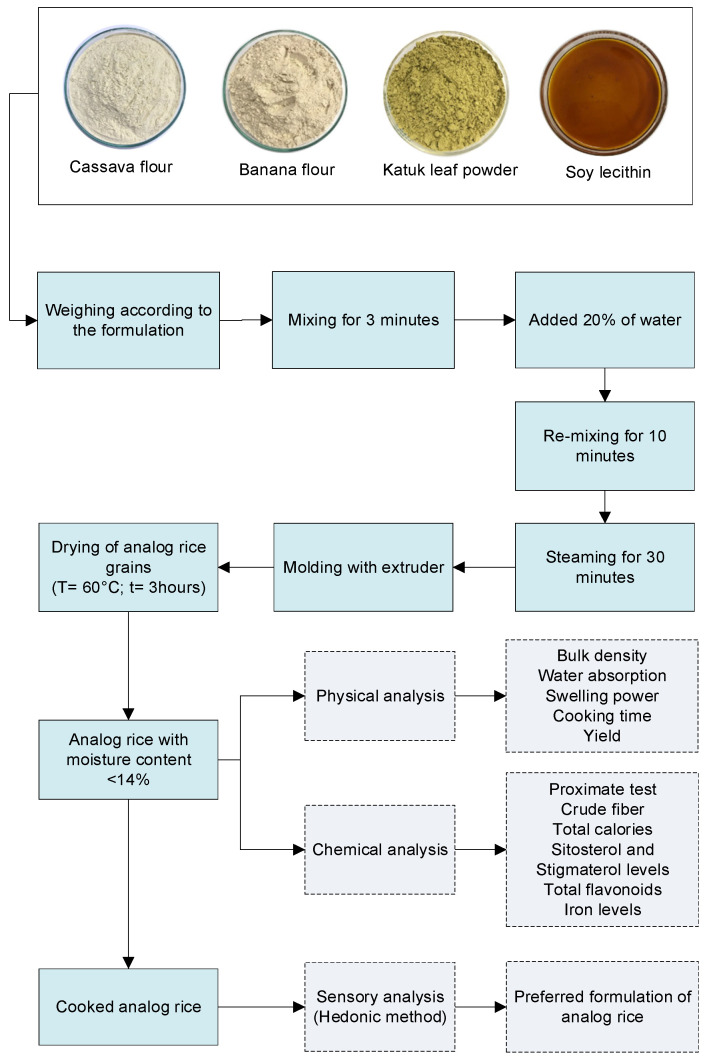
Detailed illustration of the experimental method.

**Figure 2 foods-13-01438-f002:**
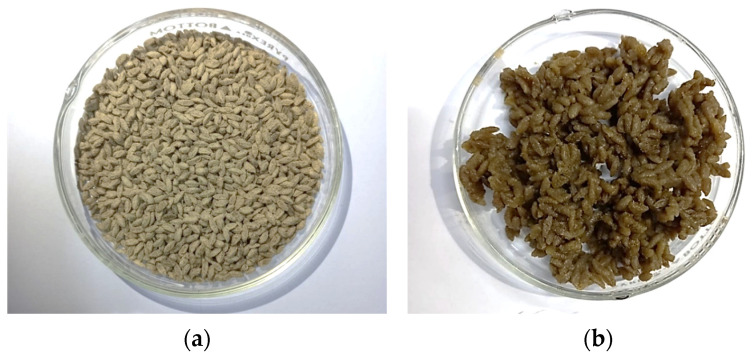
Preferred analog rice with a ratio of cassava flour and banana flour 80%:20%. (**a**) Raw analog rice, (**b**) cooked analog rice.

**Figure 3 foods-13-01438-f003:**
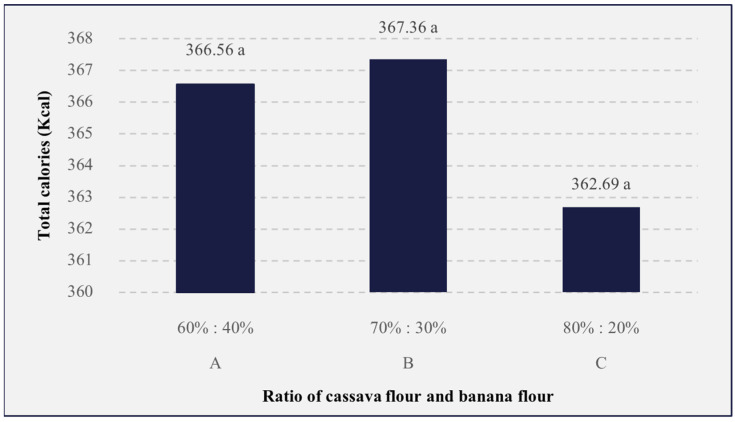
The effects of the cassava flour and banana flour ratio on the total calories of analog rice. A, B, and C are the sample codes stated on Table 1. Mean values with the same letters are insignificantly different (*p* > 0.05).

**Figure 4 foods-13-01438-f004:**
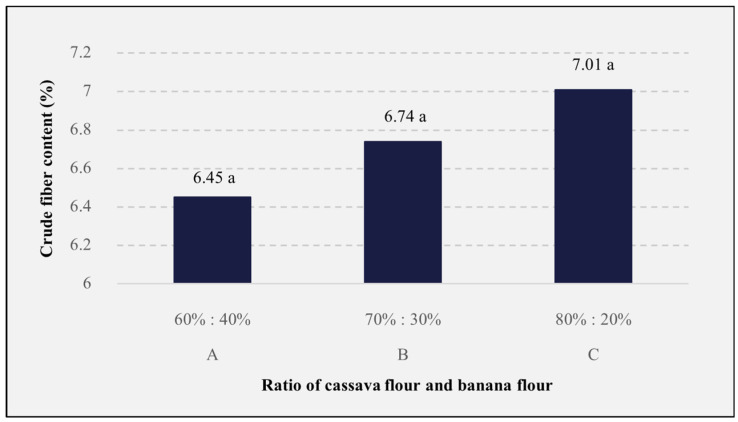
The effects of the cassava flour and banana flour ratio on the crude fiber content of analog rice. A, B, and C are the sample codes stated on Table 1. Mean values with the same letters are insignificantly different (*p* > 0.05).

**Figure 5 foods-13-01438-f005:**
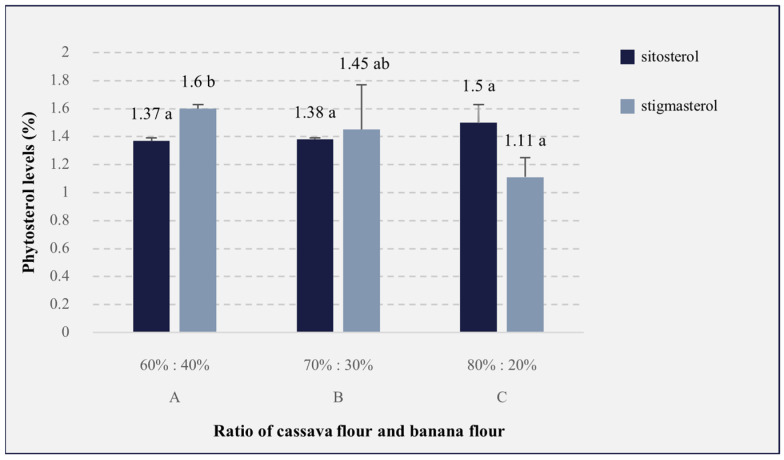
The effects of the cassava flour and banana flour ratio on the phytosterol level of analog rice. A, B, and C are the sample codes stated on Table 1. Mean values with different letters are significantly different (*p* < 0.05).

**Figure 6 foods-13-01438-f006:**
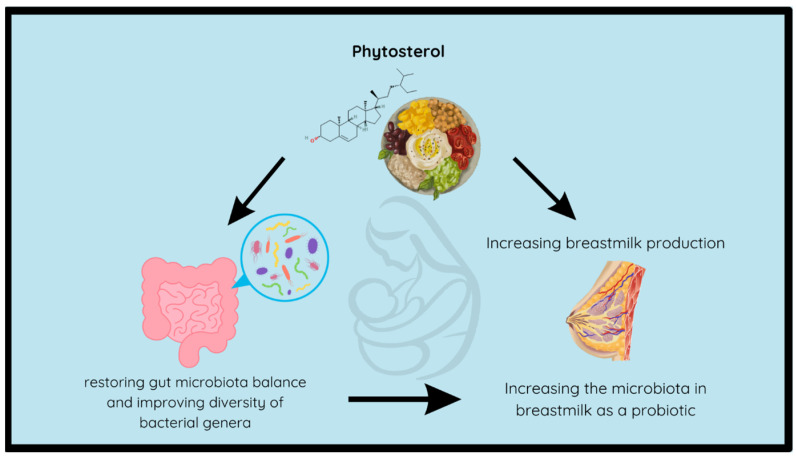
The schema of phytosterol in increasing breast milk production and quality in lactating women.

**Figure 7 foods-13-01438-f007:**
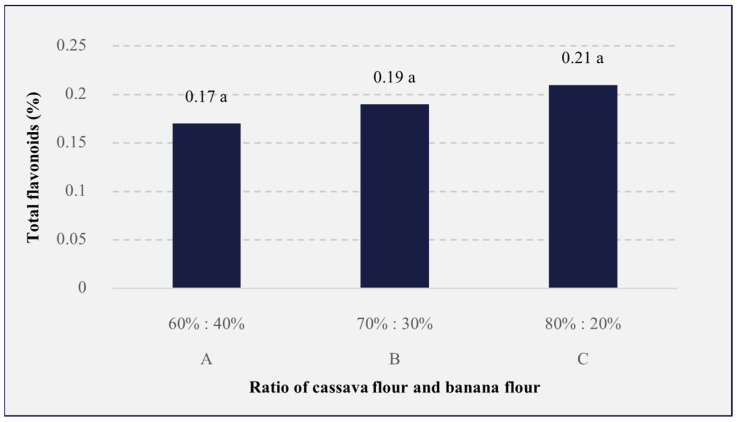
The effects of the cassava flour and banana flour ratio on the total flavonoids in analog rice. A, B, and C are the sample codes stated on Table 1. Mean values with the same letters are insignificantly different (*p* > 0.05).

**Figure 8 foods-13-01438-f008:**
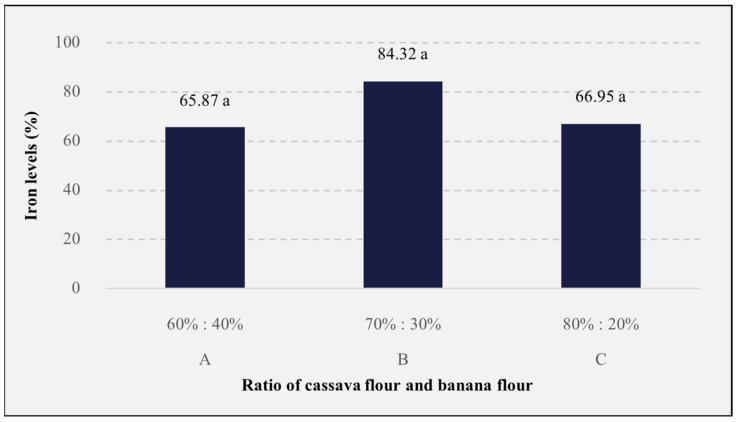
The effects of the cassava flour and banana flour ratio on the iron levels of analog rice. A, B, and C are the sample codes stated on Table 1. Mean values with the same letters are insignificantly different (*p* > 0.05).

**Figure 9 foods-13-01438-f009:**
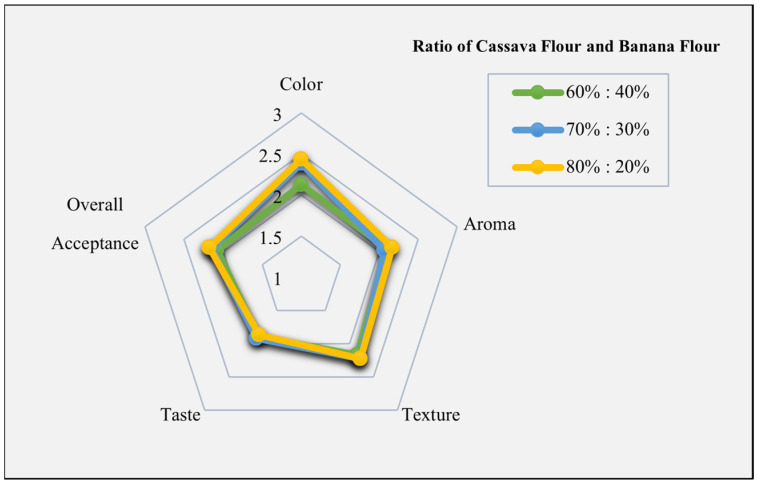
The radar chart of sensory analysis on the average scores of three analog rice formulations of different ratios of cassava and banana flour.

**Table 1 foods-13-01438-t001:** Analog rice formulation of cassava and banana flours ratios. A: 60%:40%, B: 70%:30%, C: 80%:20%.

Sample Code	Cassava Flour		Banana Flour		Katuk Leaf Powder	Soy Lecithin	Water	Total
	g	% *	g	% *	% *	% *	% *	% *
A	46.20	60	30.80	40	3	0	20	100
B	53.72	70	22.83	30	3	0.25	20	100
C	61.20	80	15.30	20	3	0.50	20	100

* Percentages based on total flour weight.

**Table 2 foods-13-01438-t002:** Results of physical analyses of analog rice.

Physical Properties	Ratio of Cassava and Banana Flours		
	A (60%:40%)	B (70%:30%)	C (80%:20%)
Bulk Density (g/mL)	0.57 ± 0.02 a	0.56 ± 0.01 a	0.55 ± 0.02 a
Water Absorption (%)	76.54 ± 1.78 a	75.08 ± 1.53 a	77.27 ± 1.15 a
Swelling Power (%)	29.45 ± 5.17 a	27.93 ± 5.01 a	31.64 ± 5.06 a
Cooking Time (minutes)	14.15 ± 0.09 a	14.06 ± 0.07 a	14.07 ± 0.03 a
Yield (%)	58.58 ± 9.25 a	61.73 ± 5.51 a	67.67 ± 1.91 a

Mean values with the same letters are insignificantly different (*p* > 0.05).

**Table 3 foods-13-01438-t003:** Proximate results of analyses of analog rice.

Proximate Properties (%)	Ratio of Cassava and Banana Flours		
	A (60%:40%)	B (70%:30%)	C (80%:20%)
Moisture content	7.15 ± 0.21 a	7.36 ± 0.69 a	7.61± 0.44 a
Ash content	2.20 ± 0.04 a	1.54 ± 1.29 a	2.32 ± 0.10 a
Protein content	2.77 ± 0.15 a	2.76 ± 0.12 a	3.07 ± 0.21 a
Fat content	0.80 ± 0.02 a	0.60 ± 0.07 b	0.48 ± 0.06 c
Carbohydrate content	87.08 ± 0.29 a	87.73 ± 1.88 a	86.53 ± 0.74 a

Mean values with different letters are significantly different (*p* < 0.05).

## Data Availability

The original contributions presented in the study are included in the article, further inquiries can be directed to the corresponding author.

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
