# Peer review of "Development of Analog Rice Made from Cassava and Banana with the Addition of Katuk Leaf (Sauropus androgynous L. Merr.) and Soy Lecithin for Lactating Women"

_foods, 2024, doi:10.3390/foods13101438_

Round 1
Reviewer 1 Report
Comments and Suggestions for Authors
The manuscript entitled "Development of Analog Rice Made from Cassava and Banana with the addition of Katuk Leaf (Sauropus androgynous L. Merr.) and soy lecithin for lactating women" aimed to determine the best formulation of analog rice made from cassava flour, banana flour, Katuk leaf powder and soy lecithin that was sensory acceptable, in a shape resembling rice. The results indicated that the preferred formulation based on sensory analysis was obtained. The work is interesting, and some revisions should be made as follows:
(1) L48: “other type of food”, which type?
(2) The Introduction needs to be concise.
(3) The formulation was only based on sensory analysis. It is better to add some experiments on texture, viscoelasticity, flavors and digestive to support your results.
(4) It is better to add vitro and vivo tests to prove the functionality of analog rice.
Author Response
Thank you very much for the suggestions and corrections of our manuscript. We have revised some of that but some we could not do because the research has been completed. Our response to the reviewer conveys the explanation and reasons for this. These suggestions will be good input for our future research. The complete response can be seen in the attached file.

Reviewer 2 Report
Comments and Suggestions for Authors
The manuscript called "Development of Analog Rice Made from Cassava and Banana with The Addition of Katuk Leaf (Sauropus androgynous L. Merr.) and Soy Lecithin for Lactating Women” consists of valuable, up-to-date research, however, the authors have carelessly prepared its record, which significantly impairs the reception of the information contained. Below are some comments that should be corrected before it is published:
Material and Method:
Line 84-85:
When referring to the work of other authors and adding that modifications have been made to the recipe of rice analogs, it should be described in detail and indicate what influenced the proportions and composition of ingredients.
Line 89-103
The production process should be described precisely and the whole thing presented in such a way that no one has any doubts about how the product was created, especially when you refer to the literature where your modifications were introduced.
Line 100
Is it appropriate to use the word "printed" when referring to extrusion? it would be better to use the term extrusion-cooking using an extruder (please provide the model and manufacturer of the extruder, and the die used).
Line 102
please remove spaces
Line 103
please describe how humidity was measured
Figure 1
Please be consistent, if parameters are provided, please enter them for each stage.
Line 107-113
The methodology should be described in such a way that it can be reproduced.
Table 2-3, figure 3-8
When we refer to Duncan's tests, we should show significant differences in the table "using letters, e.g.", even if there are none.
Line 207-269
The chemical composition should also be determined in the raw material because the composition of the analog influences the values of the final product
Line 236-241
Introducing the results, the aspect of the impact of the extrusion process, into the discussion shows that it is necessary to describe the value of the chemical composition of the basic raw material to discuss it.
Line 254-256
This sentence does not make sense (if we do not provide the fat content of the input material.
FIGURE 3-8
"ratio correlation" - such a signature is inadequate to what is shown on the chart. By "ratio correlation" we mean correlations, e.g. Pearson's, and you did not do this.
Generally, despite the rich literature, the entire discussion is very poor and should be expanded and discussed with current research.
Author Response
Thank you very much for taking the time to review and for giving such valuable suggestions and comments on our manuscript. We carefully studied your comments and have made the necessary corrections to the manuscript accordingly. We have incorporated the suggested changes and highlighted them in red within the manuscript. In accordance with your comment on Materials and Methods, we have provided a comprehensive explanation of the formulation and process involved in producing and analyzing analog rice. Please refer to the point-by-point response to the comments and concerns below, where we have addressed your insightful comments. All page numbers refer to the revised manuscript file with tracked changes. Thank you again for your time and consideration.
